# Therapeutic normal IgG intravenous immunoglobulin activates Wnt-β-catenin pathway in dendritic cells

Anupama Karnam [1], Naresh Rambabu[1], Mrinmoy Das [1], Melissa Bou-Jaoudeh [1], Sandrine Delignat[1], Fabian Käsermann [2], Sébastien Lacroix-Desmazes[1], Srini V. Kaveri[1] & Jagadeesh Bayry [1✉]

Therapeutic normal IgG intravenous immunoglobulin (IVIG) is a well-established first-line immunotherapy for many autoimmune and inflammatory diseases. Though several mechanisms have been proposed for the anti-inflammatory actions of IVIG, associated signaling pathways are not well studied. As β-catenin, the central component of the canonical Wnt pathway, plays an important role in imparting tolerogenic properties to dendritic cells (DCs) and in reducing inflammation, we explored whether IVIG induces the β-catenin pathway to exert anti-inflammatory effects. We show that IVIG in an IgG-sialylation independent manner activates β-catenin in human DCs along with upregulation of Wnt5a secretion. Mechanistically, β-catenin activation by IVIG requires intact IgG and LRP5/6 co-receptors, but FcγRIIA and Syk are not implicated. Despite induction of β-catenin, this pathway is dispensable for anti-inflammatory actions of IVIG in vitro and for mediating the protection against experimental autoimmune encephalomyelitis in vivo in mice, and reciprocal regulation of effector Th17/Th1 and regulatory T cells.

[1] Institut National de la Santé et de la Recherche Médicale, Centre de Recherche des Cordeliers, Sorbonne Université, Université de Paris, 15 rue de l'Ecole de Médicine, F-75006 Paris, France. [2] CSL Behring, Research, CSL Biologics Research Center, 3014 Bern, Switzerland. ✉email: jagadeesh.bayry@crc.jussieu.fr

ntravenous and subcutaneous immunoglobulin (IVIG/SCIG) are the therapeutic normal IgG preparations obtained from the pooled plasma of many thousands of healthy donors. In addition to its application in the replacement therapy of primary and secondary immune deficiencies, high-dose (1–2 g/kg) IVIG is used for the immunotherapy of large number of systemic auto-immune and inflammatory diseases like immune thrombocytopenic purpura, Kawasaki disease, Guillain-Barré syndrome, transplantation, inflammatory myopathies, and many others. In addition, IVIG therapy has been explored over 100 different pathologies as an off-label manner[1,2].

Based on the studies performed both in human and experimental models, several mechanisms have been proposed for IVIG. The current evidence suggests that IVIG suppresses the activation and function of various innate immune cells (like dendritic cells (DCs), neutrophils, monocytes, macrophages) and effector T and B cells. On the other hand, IVIG promotes the anti-inflammatory processes such as enhancing the tolerogenic properties of antigen presenting cells, increasing the secretion of anti-inflammatory molecules like IL-1RA and IL-10, and expansion of regulatory T cells (Tregs)[3–6]. The signaling pathways implicated in anti-inflammatory actions of IVIG are however not completely known and are the subjects of intense research.

β-catenin, the central component of the canonical Wnt signaling pathway plays a major role in balancing the immune aggression verses tolerance[7–13]. In addition to nonimmune cells, β-catenin is expressed in all immune cells including DCs, macrophages and T cells, and regulate their functions[10,14–16]. In fact, conditioning of DCs with Wnt ligands promotes the expansion of Tregs and limit the expansion of Th1/Th17 cells[12,17]. In experimental autoimmune encephalomyelitis (EAE) model, lack of β-catenin in DCs leads to increased severity of the disease[12,17,18]. Conversely, treatment with β-catenin agonists protects the mice from EAE with reduced frequency of IFN-γ and IL-17-expressing cells[12].

It is interesting to note that IVIG has been reported to suppress the inflammatory cytokines in DCs and reciprocally regulate Tregs and effector Th1/Th17 responses in vitro, in vivo in EAE model, and in treated autoimmune patients[19–33]. In view of reports on the central role of β-catenin/Wnt pathway in mediating these anti-inflammatory effects, we hypothesized that IVIG induces the activation of β-catenin/Wnt pathway to exert anti-inflammatory effects.

In the present report, we show that IVIG indeed activates the β-catenin pathway along with Wnt5a secretion in human DCs in a sialylation independent manner. Mechanistically, β-catenin activation by IVIG requires low-density lipoprotein receptor related proteins (LRP) 5 and 6 co-receptors but FcγRII and Syk pathway are not implicated. However, despite induction of β-catenin/Wnt signaling, we found that this pathway is dispensable for the anti-inflammatory action of IVIG both in vitro, and in vivo in mediating the protection against EAE in mice and reciprocal regulation of effector Th17/Th1 and Tregs.

## Results

**IVIG stimulates β-catenin signaling in DCs.** Glycogen synthase kinase-3 beta (GSK-3β) acts as a negative regulator of β-catenin pathway by phosphorylating β-catenin at Ser33, Ser37 and Thr41, and directing it to proteasomal degradation[34]. Non-phosphorylated forms of β-catenin are functionally active and translocate to nucleus and regulate the expression of target genes. In view of critical role of DCs in mediating both immune response and tolerance, we first investigated whether IVIG activates β-catenin signaling in these innate immune cells. Monocyte-derived DCs were treated with 25 mg/ml of IVIG (equivalent of

concentrations of infused IgG reaches in the blood of auto-immune patients immediately following IVIG immunotherapy) for 24 h and activation of β-catenin signaling was analyzed by immunoblotting for active form of β-catenin (non-phospho β-catenin).

We found that IVIG significantly increased the levels of active β-catenin, which is not phosphorylated at Ser33, Ser37, and Thr41 residues (Fig. 1a). On the other hand, equimolar concentrations of human serum albumin (HSA), an irrelevant protein control for IVIG, did not alter the basal levels of active β-catenin and were similar to untreated cells.

We have also assessed the levels of triple phosphorylated β-catenin (Ser33, Ser37, and Thr41). Consistent with the data on active β-catenin (Fig. 1a), IVIG significantly decreased the phospho-β-catenin (Fig. 1b). The effect of IVIG was dose-dependent. Furthermore, IVIG promoted the translocation of β-catenin into the nucleus, thus validating the activation of β-catenin pathway in IVIG-treated DCs (Fig. 1c).

We explored if increase in the levels of active β-catenin by IVIG was due to enhancement in the β-catenin levels. For this purpose, we first performed *CTNNB1* gene (that encodes β-catenin) expression analyses by quantitative reverse transcription-polymerase chain reaction (RT-PCR) and found that IVIG significantly enhanced the *CTNNB1* mRNA in DCs (Fig. 1d). However, analyses of β-catenin protein by western blot did not reveal significant differences between various experimental conditions (Fig. 1e). These data thus imply that IVIG-mediated increase in the levels of active β-catenin is not because of enhancement in the β-catenin protein levels.

As GSK-3β negatively regulates active β-catenin, we wondered if the positive effect of IVIG on β-catenin is associated with concomitant inhibition of GSK-3β. Of note, GSK-3β protein was significantly downregulated by IVIG (Fig. 2), thus signifying that IVIG-induced activation of β-catenin is coupled with reduced GSK-3β levels. Together, these results show that IVIG activates β-catenin signaling in human DCs.

**IVIG activates β-catenin in DCs in an IgG-sialylation independent manner.** We then explored the mechanisms of activation of β-catenin in DCs by IVIG. Several reports have shown the importance of terminal α-(2,6) sialic acid linkages of Fc region in mediating the anti-inflammatory actions of IVIG[31,35–37]. Therefore, we examined if IVIG-induced activation of β-catenin is dependent on the sialylation content of IgG. IVIG was desialylated by treatment with recombinant neuraminidase. The sialylation content in neuraminidase-treated IVIG was <2.5 µg/25 mg of IgG as assessed by reverse phase high performance liquid chromatography[38,39]. Lectin-blot analyses also confirmed the desialylation of IVIG[38]. By using desialylated IVIG, we found that IVIG-induced activation of β-catenin in DCs is independent of sialylation status of IVIG (Fig. 3). In fact, desialylated IVIG-induced activation of β-catenin similar to that of native IVIG.

**β-catenin activation by IVIG does not imply C-type lectin and FcγRIIA receptors.** C-type lectin receptors like Dectin-1 and FcγR-mediated signaling have been reported to induce β-catenin activation and its stabilization[40,41]. To investigate if IVIG induces β-catenin activation via C-type lectin receptors, we treated the DCs with $Ca^{2+}$ chelating agent ethylenediaminetetraacetic acid (EDTA) followed by IVIG treatment. However, IVIG-induced β-catenin activation remained intact upon treatment of DCs with EDTA (Fig. 4a), implying that IVIG stimulates β-catenin activation in a $Ca^{2+}$-independent manner.

Monocyte-derived DCs mainly express FcγRII, while FcγRIII and FcγRI are either absent or marginal[42,43]. However,

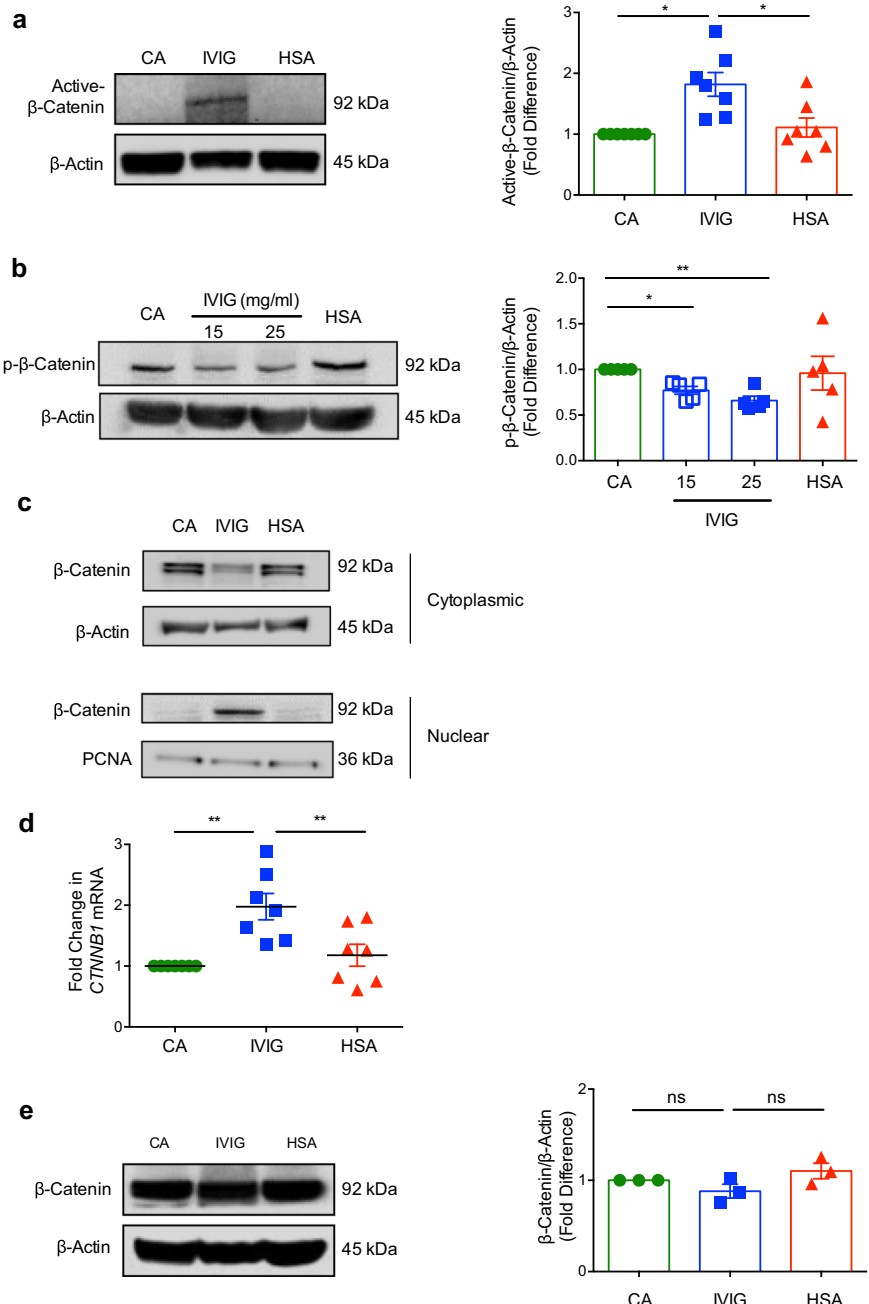

**Fig. 1 Intravenous immunoglobulin (IVIG) stimulates β-catenin signaling in dendritic cells.** Monocyte-derived DCs (0.5 million cells/ml) were cultured either alone (cells alone, CA), with IVIG (15 or 25 mg/ml) or with an equimolar concentration of human serum albumin (HSA) for 24 h. **a** Immunoblot analysis of active-β-catenin ($n = 7$ donors) and **b** Immunoblot analysis of p-β-catenin ($n = 5$ donors). Representative immunoblots and densitometric analyses of the blots (mean ± SEM) are presented. β-Actin was used as a loading control. Images are cropped for the presentation. **c** Investigation of nuclear translocation of β-catenin protein by immunoblot. Representative immunoblot from DCs of two donors is presented. **d** DCs were treated with IVIG or HSA for 12 h and the mRNA levels of *CTNNB1* was analyzed by quantitative real-time RT-PCR (mean ± SEM, $n = 7$ donors). **e** Changes in the levels β-catenin protein in IVIG-treated DCs ($n = 3$) as analyzed by immunoblot. *$P < 0.05$; **$P < 0.01$; ns, not significant; as determined by one-way ANOVA followed by Tukey's multiple comparisons test.

FcγRIIA-blockade with monoclonal antibodies did not affect IVIG-induced β-catenin activation in DCs (Fig. 4b). As C-type lectin receptors and activating FcγR mediate signaling via Syk pathway[44,45], to rule out the implication of both the receptors, we employed Syk inhibitor in the experiments. DCs were pre-treated with Syk inhibitor followed by culture with IVIG. Consistent with the previous results, Syk inhibition had no effect on the ability of IVIG to induce β-catenin activation (Fig. 4c).

**Intact IgG is mandatory for the β-catenin activation by IVIG.** We next aimed at dissecting whether induction of β-catenin activation by IVIG implicates F(ab′)$_2$ or Fc fragments. DCs were treated with IVIG (25 mg/ml) and equimolar concentrations of F(ab′)$_2$ or Fc fragments. In line with the data on the lack of implications of sialylation of IgG, Syk and FcγRIIA in mediating β-catenin activation by IVIG, Fc fragments of IVIG failed to induce β-catenin activation (Fig. 5a).

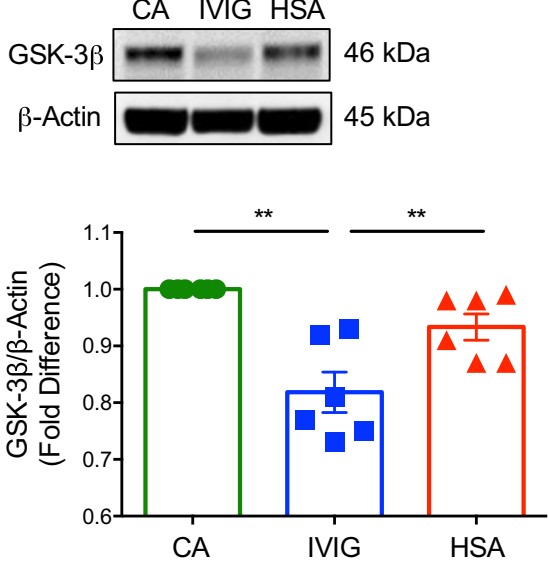

**Fig. 2 Intravenous immunoglobulin (IVIG)-induced changes in the expression of GSK-3β in dendritic cells.** Monocyte-derived DCs (0.5 million cells/ml) were cultured either alone (cells alone, CA), with IVIG (25 mg/ml) or with an equimolar concentration of human serum albumin (HSA) for 24 h. Immunoblot analysis of GSK-3β was performed. Representative immunoblot and densitometric analyses of the blots (mean ± SEM, $n = 6$ donors) are presented. β-Actin was used as a loading control. **$P < 0.01$; as determined by one-way ANOVA followed by Tukey's multiple comparisons test.

These results raised the prospect that IVIG might induce β-catenin activation via F(ab′)$_2$ fragments. However, we did not observe β-catenin activation even with F(ab′)$_2$ fragments of IVIG (Fig. 5b). Together our data indicate that intact IgG is mandatory for the β-catenin activation by IVIG.

**Wnt5a and LRP5/6 interaction is critical for β-catenin activation by IVIG.** We then explored if Wnt-receptor interaction is required for the activation of β-catenin by IVIG. Signaling by Wnt ligands upon binding to transmembrane Frizzled (Fz) receptor and its co-receptors, LRP5/6 leads to the transduction of signals and accumulation of β-catenin in the cytoplasm by preventing GSK-3β-mediated phosphorylation and proteosomal degradation of β-catenin. Eventually, β-catenin translocate into the nucleus to activate Wnt target genes (canonical Wnt/β-catenin pathway)[8,9,14,46,47].

We first screened for the expression of canonical ligands by quantitative RT-PCR. Levels of mRNA for canonical ligands like WNT3A, WNT1, and WNT10A were significantly upregulated upon 12 h of IVIG treatment of DCs (Fig. 6a).

Wnt ligands are also known to mediate noncanonical signaling pathway independent of β-catenin to regulate the cytoskeleton and intracellular calcium levels[8,48]. We hence investigated the expression of several non-canonical ligands-induced in IVIG-treated DCs. Interestingly, IVIG also enhanced the expression of several Wnt ligands implicated in non-canonical pathway like WNT5A, WNT7A, and WNT7B (Fig. 6b) though significant expression was observed only with WNT5A.

These data encouraged us to assess the secretion of prototype canonical ligand protein Wnt3a and noncanonical ligand protein Wnt5a in the culture supernatants. In contrast to transcript analyses, however, Wnt3a was not secreted in IVIG-treated DCs. Only Wnt5a was significantly secreted by these

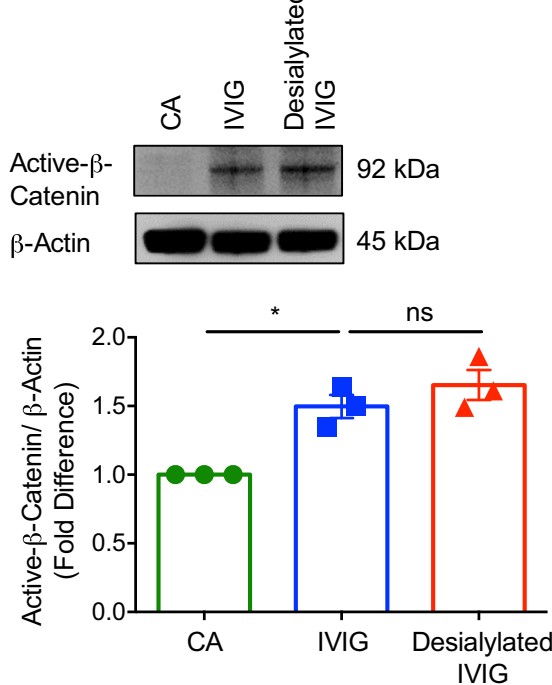

**Fig. 3 Intravenous immunoglobulin (IVIG) activates β-catenin in human dendritic cells in an IgG-sialylation independent manner.** DCs (0.5 million cells/ml) were cultured either alone (cells alone, CA), treated with IVIG (25 mg/ml) or with desialylated IVIG (25 mg/ml) for 24 h. Activation of β-catenin was analyzed by immunoblotting. β-Actin was used as a loading control. Representative immunoblot and densitometric analyses of the blots (mean ± SEM, $n = 3$ donors) are presented. *$P < 0.05$; ns, not significant; as determined by one-way ANOVA followed by Tukey's multiple comparisons test.

DCs (over 10 ng/ml) (Fig. 6c). These results thus highlight the possible involvement of noncanonical Wnt5a in the IVIG-induced activation of β-catenin signaling in human DCs.

Though Wnt5a is typically associated with noncanonical Wnt signaling, it could activate Wnt/β-catenin pathway in the presence of Fz4 and LRP5[49,50]. Therefore, to unequivocally prove that IVIG triggered β-catenin activation in DCs through LRP5-mediated signaling of Wnt5a, we resorted to silence LRP5 gene by siRNA method. Although LRP6 might not be binding to Wnt5a, to prevent the binding of other canonical Wnt ligands, we decided to knock-down both LRP5 and 6 co-receptor genes to ensure complete inhibition of β-catenin activation signals. In line with our hypothesis, the ability of IVIG to activate β-catenin was compromised when DCs are treated with siRNA against LRP5/6 (Fig. 7).

Together, these data imply that IVIG-induced activation of β-catenin pathway in DCs is dependent on the Wnt5a- LRP5/6 co-receptor interaction.

**β-catenin is dispensable for the anti inflammatory effects of IVIG on DCs.** The critical question was whether β-catenin signaling is important for the anti-inflammatory actions of IVIG. We addressed this by using FH535, a small molecule inhibitor of Wnt/β-catenin signaling[51]. We first examined the inhibitory effect of FH535 on the levels of IVIG-induced active β-catenin (non-phospho β-catenin) in DCs. As shown in Fig. 8a, FH535 significantly reduced the levels of active β-catenin thus confirming that FH535 is functional[52].

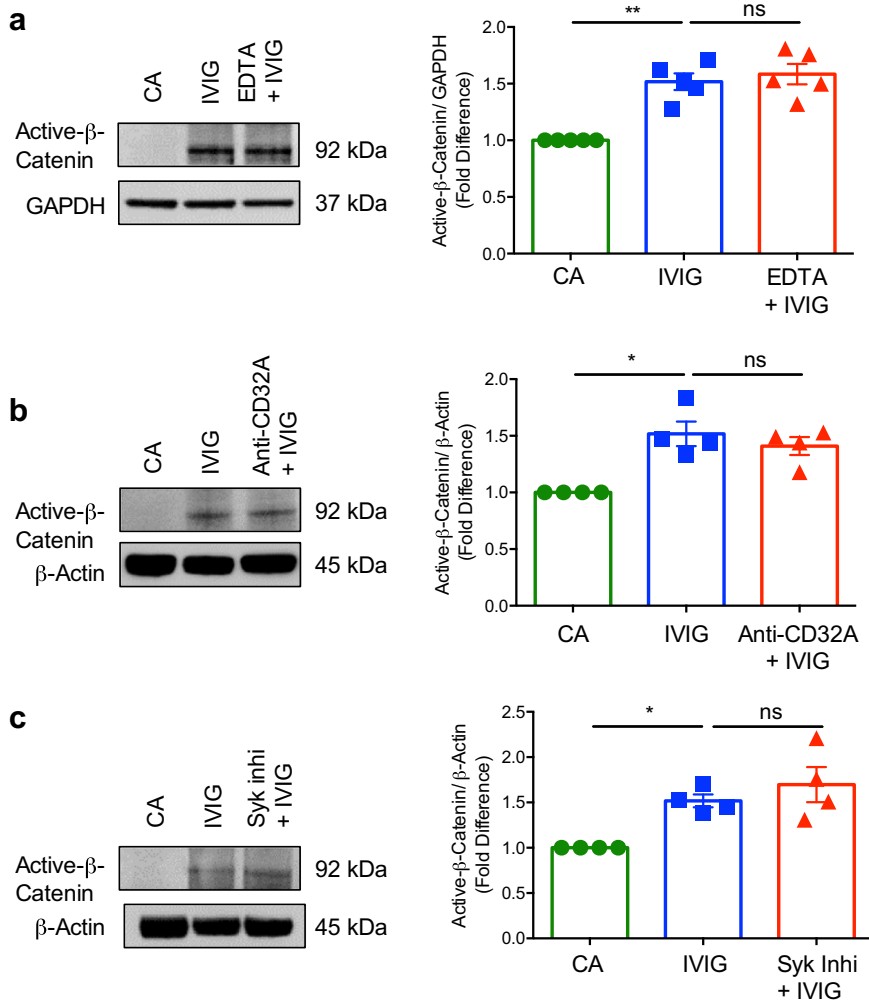

**Fig. 4 Lack of implications of C-type lectin receptors and activating FcγRIIA in mediating β-catenin activation by intravenous immunoglobulin (IVIG) in dendritic cells.** DCs were pre-treated with (**a**) EDTA or (**b**) anti-CD32A MAb or (**c**) Syk inhibitor for one hour followed by treatment with IVIG for 24 h. Representative immunoblots and densitometric analyses of the blots (mean ± SEM, $n = 4$–5 donors) are presented. GAPDH (**a**) or β-Actin (**b** and **c**) were used as loading controls. *$P < 0.05$; **$P < 0.01$; ns, not significant; as determined by one-way ANOVA followed by Tukey's multiple comparisons test. CA cells alone.

Previous reports have shown that β-catenin pathway regulates the inflammatory cytokine response in toll-like receptor 4 (TLR4)-activated DCs[53,54]. Therefore, by using TLR4-activated DCs (lipopolysaccharide, LPS stimulation), we investigated if β-catenin signaling is implicated in the anti-inflammatory actions of IVIG. As depicted in Fig. 8b, IVIG reduced the production of IL-6 and IL-8 in activated DCs, thus validating the suppressive effects of IVIG on DCs[19]. However, inhibition of Wnt/β-catenin signaling by FH535 had no repercussion on the inhibitory actions of IVIG on the production of these inflammatory cytokines. These results suggest that β-catenin though induced by IVIG, this pathway is dispensable for the anti-inflammatory action of IVIG.

**β-catenin is dispensable for the anti inflammatory effects of IVIG in vivo.** To validate the role of β-catenin signaling in the anti-inflammatory actions of IVIG in vivo, we resorted to EAE model. Previous reports have shown that IVIG protects the mice from the disease and is associated with the peripheral expansion of Tregs and suppression of pathogenic Th1 and Th17 cells[24,26].

We first wanted to confirm that IVIG induces activation of β-catenin in mouse immune cells. Culturing of splenocytes from C57/BL6 mice with IVIG for 24 h induced active-β-catenin while HSA had no such effect (Fig. 9a). Thus, irrespective of species (human or mouse), IVIG stimulates β-catenin signaling.

Furthermore, inhibition of β-catenin signaling in vivo in EAE model by daily injection of β-catenin antagonist (FH535) along with IVIG did not compromise the protective effects of IVIG. The clinical scores were similar to IVIG injected in combination with dimethyl sulfoxide (DMSO, solvent control) (Fig. 9b). The control mice developed the signs of EAE by day 12, whereas the onset of the disease was delayed in both IVIG + solvent control and IVIG + β-catenin antagonist FH535 groups. Further, the severity of disease remained mild irrespective of mice received inhibitor or not (Fig. 9b).

We analyzed the various subsets of CD4[+] T cells in the spleen of mice on the day of onset of disease. Confirming the previous reports, IVIG suppressed both Th1 and Th17 cells and reciprocally enhanced Tregs (Fig. 9c, d). Importantly, β-catenin antagonist had no effect on the IVIG-mediated reciprocal regulation of Tregs and effector Th1 and Th17 cells (Fig. 9c, d).

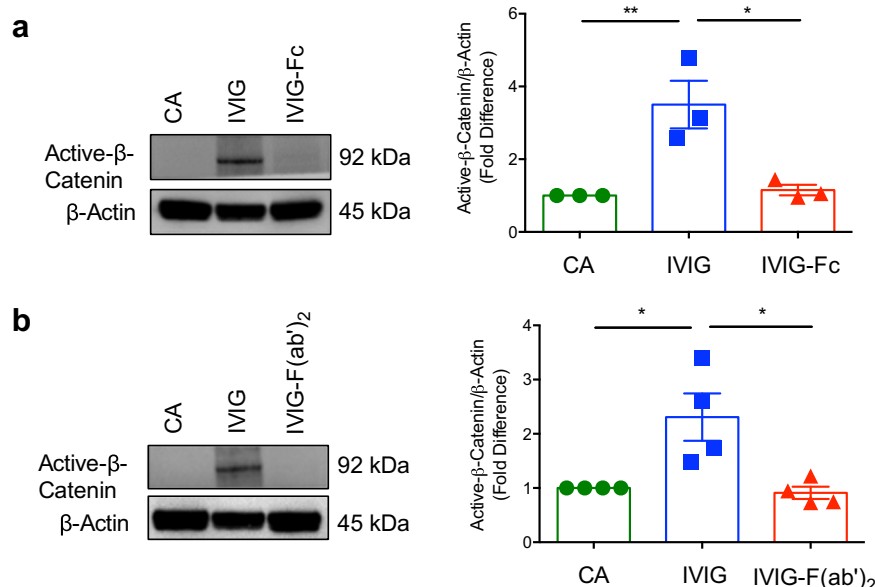

**Fig. 5 Intact IgG is mandatory for the β-catenin activation by intravenous immunoglobulin (IVIG).** DCs (0.5 million cells/ml) were cultured either alone (cells alone, CA), treated with IVIG (25 mg/ml) or equimolar concentrations of (**a**) Fc or (**b**) F(ab')$_2$ fragments of IVIG for 24 h. Activation of β-catenin was analyzed by immunoblotting. β-Actin was used as a loading control. Representative immunoblots and densitometric analyses of the blots (mean ± SEM, $n = 3$–4 donors) are presented. *$P < 0.05$; **$P < 0.01$; ns, not significant; as determined by one-way ANOVA followed by Tukey's multiple comparisons test.

Together, these data demonstrate that β-catenin signaling is dispensable for IVIG-mediated anti-inflammatory effects in vivo.

## Discussion

Despite its widespread use over the last 35 years as an immunotherapeutic agent in autoimmune and inflammatory diseases, our knowledge on the mechanisms by which IVIG benefits such a vast number of pathologies is still incomplete. IVIG interacts with various arms of the immune system and exerts anti-inflammatory effects by diverse mechanisms that appear to function in coordination. While mechanisms like saturation of neonatal Fc receptor (FcRn), neutralization of pathogenic autoantibodies and inflammatory cytokines, and complement scavenging are functioning during early phase of IVIG therapy; regulation of immune cells like innate cells (like DCs, neutrophils, monocytes, macrophages, basophils) and adaptive immune cells (T and B cells) occur during subsequent phase to ensure immune homeostasis[3,4,19,26,55–60].

DCs play a crucial role as professional antigen presenting cells in balancing the immune response and tolerance. Investigations on the mode of action of IVIG have revealed that it inhibits the activation of both human and murine DCs and the production of inflammatory cytokines[19,23,27] The ability of IVIG to suppress DC activation also had impact on the DC-mediated T cell responses and the pathogenesis of autoimmune diseases[19,20,55,61,62]. The signaling pathway(s) that are implicated in the regulation of DC functions by IVIG is not well understood. Previous reports have shown that preconditioning of human monocyte-derived DCs with IVIG impairs TLR4-mediated activation of extracellular signal–regulated kinases 1/2 (ERK1/2)[63]. Although ERK1/2 was not inhibited by IVIG in murine bone marrow-derived DCs, when these cells were stimulated with LPS along with IVIG, p38 mitogen activated protein kinase (p38MAPK) activation was suppressed[23]. Possibly, differences in the concentrations of IVIG and/or experimental conditions might have contributed to the differences observed between human and mouse DCs. Type II C-type lectin receptors appear to play an important role in transducing the signaling events by IVIG in DCs. However, the nature of the lectin receptor varies depending on the subset of DCs[64,65]. By signaling via DC-SIGN (Dendritic Cell-Specific Intercellular adhesion molecule-3-Grabbing Non-integrin), IVIG and its F(ab')$_2$ fragments induced prostaglandin E2 in human DCs by activating cyclooxygenase-2 pathway and mediated Treg expansion[26]. The IL-3-produced from these activated Tregs might license basophils to undergo activation by IVIG leading to the secretion of IL-4 and further suppression of effector Th17 and Th1 cells[56,66]. On the other hand, in the humanized DC-SIGN mice model, Fc fragments containing terminal α-(2,6) sialic acids interacted with DCs and induced IL-33[35] though in human, DC-SIGN-positive DCs failed to produce IL-33 upon exposure to IVIG[67]. Ligation of DCIR (Dendritic cell immunoreceptor) by I-VIG induced phosphorylation of immunoreceptor tyrosine-based inhibition motif (ITIM)-linked phosphatases Src homology 2 (SH2) domain-containing inositol polyphosphate 5-phosphatase-1 (SHIP-1) and SH2-containing tyrosine phosphatase-2 (SHP-2) in pulmonary CD11c$^+$ DCs of mice[68]. Thus, IVIG activates various signaling pathways in DCs to mediate anti-inflammatory actions.

Though Wnt/β-catenin pathway is generally studied for their role in embryonic development and tumorigenesis[8], several recent studies have emphasized the role of Wnt/β-catenin pathway in the regulation of systemic and mucosal inflammatory immune responses by mediating tolerogeneic properties in DCs and ensuing T cell responses[13,18,41,69–71]. β-catenin signaling in DCs suppresses inflammatory cytokines like IL-6, induces anti-inflammatory cytokines, and reciprocally regulates Treg and Th1/Th17 responses in vivo in experimental models of autoimmune and inflammatory diseases like EAE and inflammatory bowel disease[18,69]. In line with these data on the role of β-catenin in promoting tolerance in DCs, treatment of these cells with Wnt3a proteins that trigger β-catenin activation also suppress TLR-induced pro-inflammatory cytokines in DCs[17]. As both IVIG and Wnt/β-catenin pathway mediate similar anti-inflammatory actions on DCs and T cell responses, inspired us to explore if normal IgG (IVIG) targets Wnt/β-catenin pathway to exert therapeutic benefits.

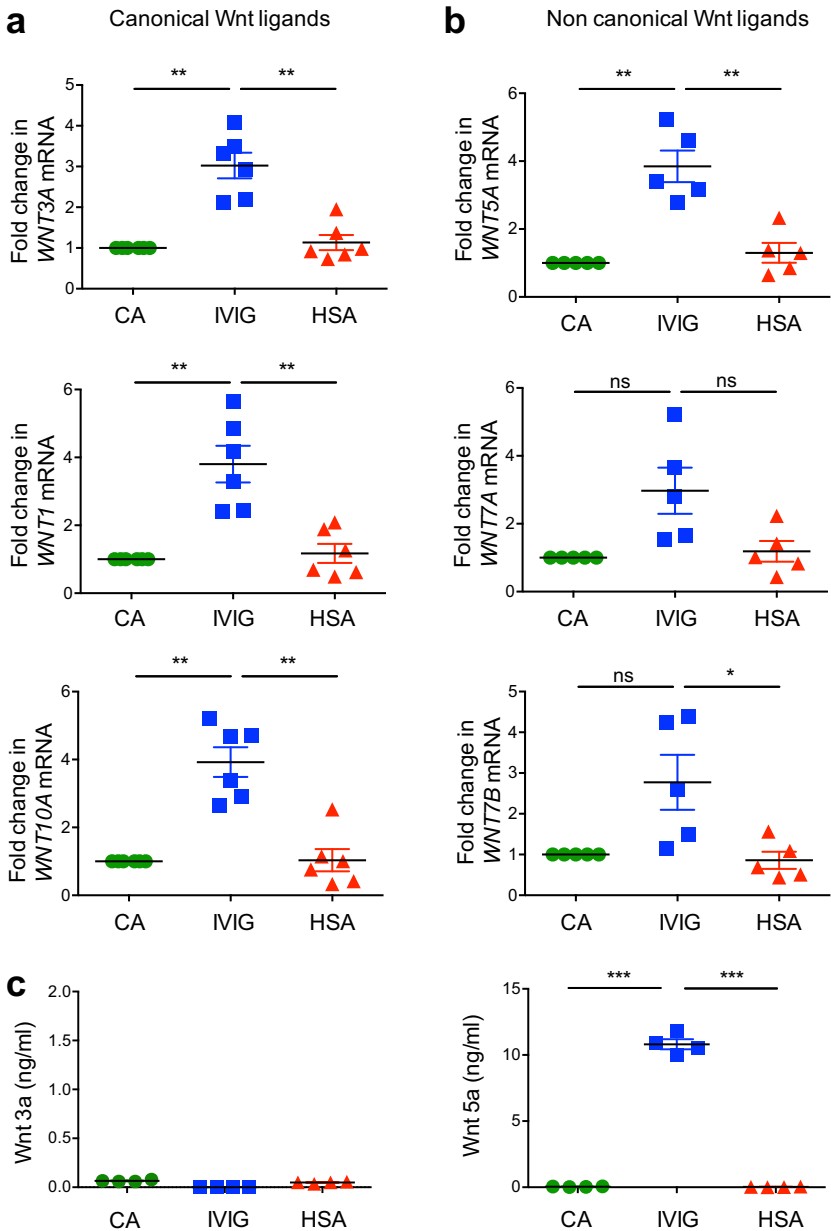

**Fig. 6 Effect of intravenous immunoglobulin (IVIG) on the induction of Wnt ligands in dendritic cells.** Monocyte-derived DCs were treated with IVIG at 25 mg/ml concentration or with equimolar concentration of HSA for 12 h. **a** The expression of various canonical Wnt ligands (mean±SEM, $n = 6$ donors) and **b** noncanonical Wnt ligands (mean±SEM, $n = 5$ donors) was analyzed by quantitative real-time RT-PCR. **c** The amount of secreted Wnt 3a and Wnt 5a in DCs treated with IVIG for 24 h (mean ± SEM, $n = 4$ donors). *$P < 0.05$; **$P < 0.01$; ***$P < 0.001$; ns, not significant; as determined by one-way ANOVA followed by Tukey's multiple comparisons test. CA cells alone, HSA human serum albumin.

In line with our proposition, IVIG induced activation of β-catenin in DCs. However, inhibition of β-catenin pathway did not compromise the ability of IVIG to suppress IL-6 and IL-8 production in DCs, suggesting that β-catenin pathway is redundant for IVIG. Our data thus affirm that IVIG is not bound by a single mechanism to exert anti-inflammatory effects. The mutually nonexclusive pathways induced by IVIG ensure that a deficiency in a particular pathway could be compensated by others. β-catenin pathway however is not only the pathway dispensable for anti-inflammatory actions of IVIG. In fact, heme oxygenase-1 (HO-1) pathway that regulates inflammatory responses by catabolizing the free heme and releasing carbon monoxide and biliverdin, is also found not essential for the anti-inflammatory effects of IVIG[72].

How does IVIG induce β-catenin activation? Mechanistically, our data imply that IVIG induces Wnt5a in DCs that signals via frizzled and LRP5 receptors to activate β-catenin pathway. Though Wnt5a is a noncanonical Wnt ligand, various reports have indicated its involvement to induce β-catenin pathway[49,50]. In the presence of Fzd4 and LRP coreceptors, Wnt5a could indeed activate β-catenin-mediated canonical pathway[49,50]. Secretion of Wnt5a in IVIG-treated DCs and abrogation of IVIG-induced β-catenin activation upon silencing of *LRP 5/6* indicate Wnt5a-induced β-catenin activation by IVIG.

Several reports have demonstrated that β-catenin could be activated by Dectin-1, TLR2, or FcγR-mediated signaling[18,40,41,73]. In our report, pre-treatment of DCs with EDTA that chelates $Ca^{2+}$ and inhibits C-type lectin receptor-mediated binding of ligands

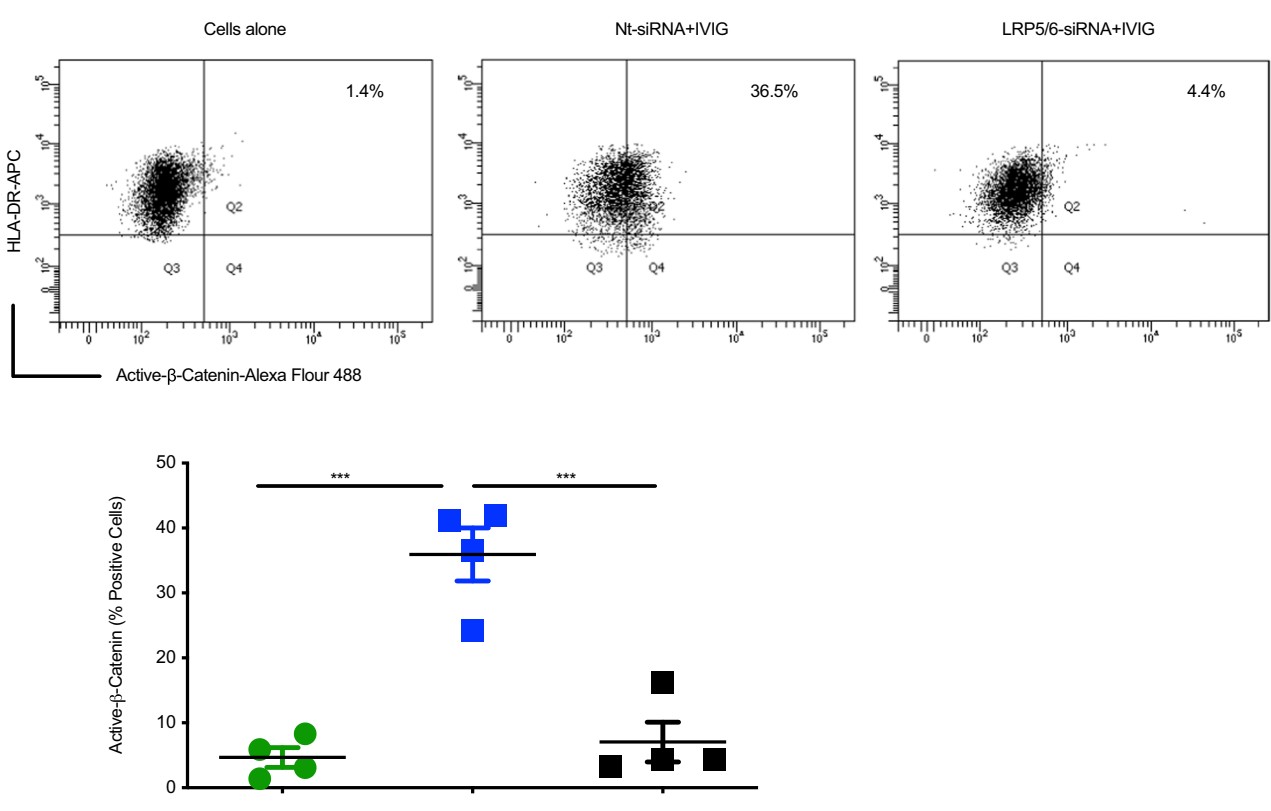

**Fig. 7 β-catenin activation by intravenous immunoglobulin requires Wnt5a-LRP5/6 co-receptors.** DCs were cultured with 1μM of Nt-siRNA or LRP5/6 siRNA for 72 h in Accell Delivery media. After 72 h, DCs were treated with IVIG followed by flow cytometric analysis for active-β-catenin expression. Representative dot plots and mean ± SEM values from the cells of four donors are presented. ***$P < 0.001$; as determined by one-way ANOVA followed by Tukey's multiple comparisons test. CA cells alone.

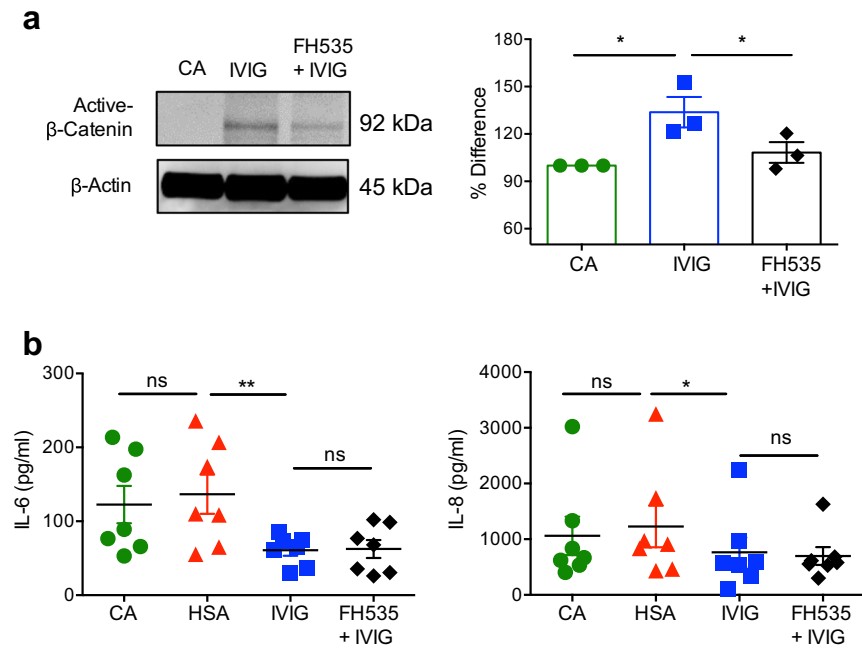

**Fig. 8 β-catenin signaling is dispensable for the anti-inflammatory actions of intravenous immunoglobulin on dendritic cells.** DCs were stimulated with LPS for 24 h. The cells were then treated with β-catenin antagonist FH535 for 2 h followed by culture with IVIG (25 mg/ml) for 24 h. **a** Levels of active β-catenin in the total cell lysate as analyzed by immunoblot. Representative immunoblot and % difference in the active β-catenin are presented (mean ± SEM, $n = 3$ donors). Images are cropped for the presentation. **b** The amounts of IL-6 and IL-8 in the cell-free culture supernatants as assessed by ELISA (mean ± SEM, $n = 7$ donors). *$P < 0.05$; **$P < 0.01$; ns, not significant; as determined by one-way ANOVA followed by Tukey's multiple comparisons test. CA cells alone treated with LPS, HSA human serum albumin.

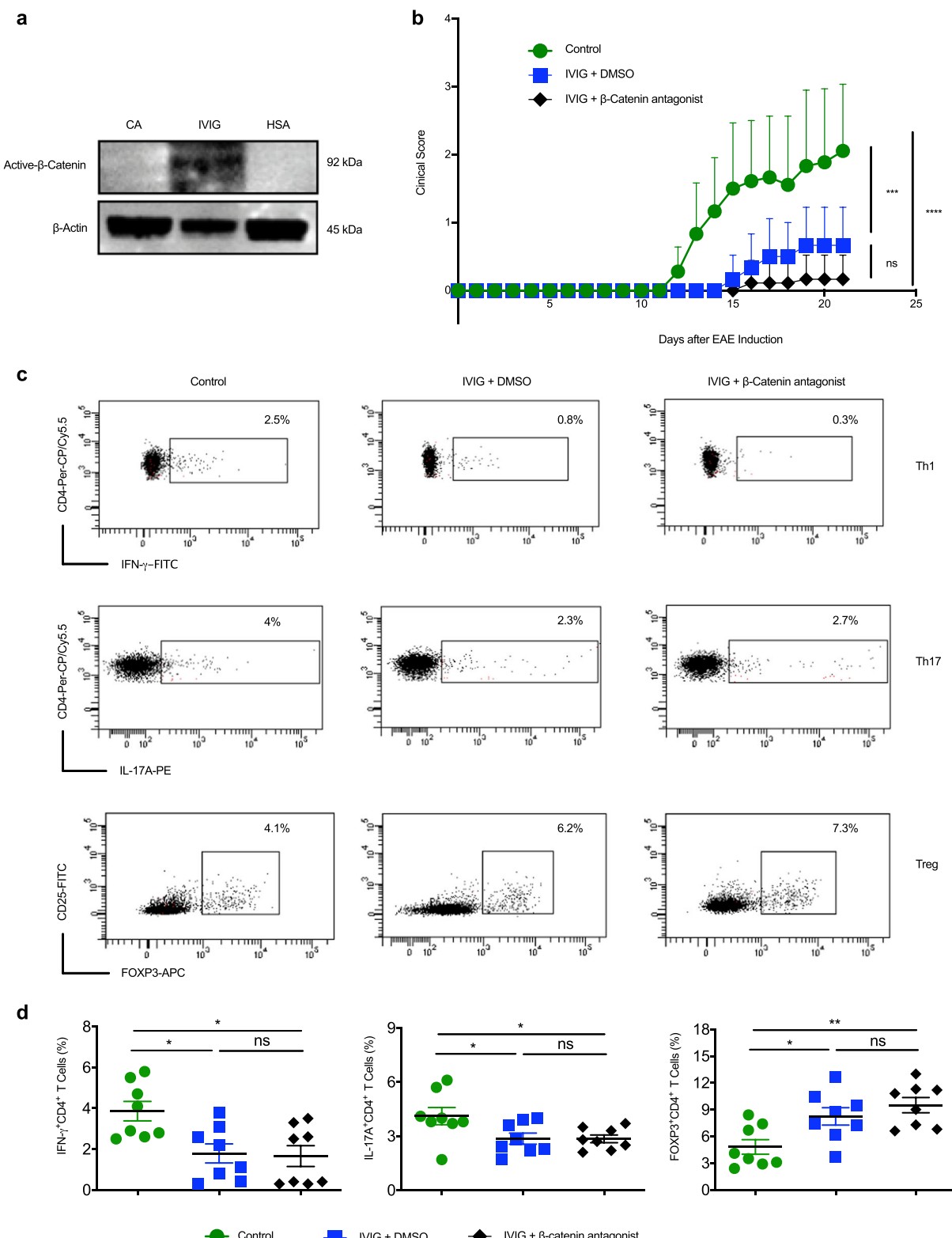

including Dectin-1, DC-SIGN and DCIR, did not prevent the ability of IVIG to induce β-catenin activation. The retention of ability to induce β-catenin activation by desialylated IVIG suggest that type II Fc receptors are also not implicated in the process[31,35,74]. The involvement of TLR2 was also precluded in view of the fact that IVIG-could induce β-catenin activation in DCs independent of TLR signaling.

We found that induction of β-catenin activation by IVIG requires intact IgG while neither Fc nor F(ab′)₂ fragments could recapitulate these functions. As human monocyte-derived DCs mainly express low affinity FcγRII, explains the inability of Fc fragments of IVIG to induce β-catenin activation. However, despite having the ability to bind DCs[19], F(ab′)₂ fragments did not promote β-catenin activation. Therefore, it is likely that

**Fig. 9 β-catenin signaling is dispensable for the IVIG-mediated protection from EAE and reciprocal regulation of Tregs and effector CD4⁺ T cells.**
**a** Splenocytes from C57/BL6 mice were treated with IVIG or HSA for 24 h followed by immunoblot analysis for the active β-catenin. Images are cropped for the presentation. **b** EAE was induced in three groups of C57/BL6 mice by immunizing with MOG$_{35-55}$ in complete Freund's adjuvant. First group of mice received no injections. Second group mice were injected with IVIG and DMSO and the third group with IVIG along with β-catenin antagonist FH535 daily until peak of the disease. Development of disease was monitored every day. Mean clinical scores from one of the two independent experiments (n = 9 mice) are presented. Error bars represent SEM. ***$p < 0.001$; ****$p < 0.0001$; ns, not significant; as determined by two-way ANOVA with Bonferroni post-$t$-test. **c** On the day of onset of clinical signs, mice were sacrificed and splenocytes were collected for the analysis of CD4⁺T cell subsets. Th1, Th17, and Treg population was analyzed by flow cytometry. Representative dot plots showing the frequency (% positive cells) of CD4⁺ IFN-γ⁺ Th1, CD4⁺ IL-17A⁺ Th17, and CD4⁺ CD25⁺ FOXP3⁺ Tregs in various groups are presented. **d** Frequency of CD4⁺ T cell subsets (Th1, Th17, and Treg) in the spleen from two independent experiments (mean ± SEM, $n = 8$ mice). *$p < 0.05$; **$p < 0.01$; ns, not significant; as determined by one-way followed by Tukey's multiple comparisons test.

cooperation between Fab and Fc regions lead to β-catenin activation by IVIG. As reported earlier, IVIG contains antibodies to various self-motifs including those antigens expressed on DCs[4–6]. Thus, binding of Fab region of IVIG to DCs might license Fc region to bind FcγRII of adjacent DCs and signal β-catenin activation in those cells. Human DCs display a balanced expression of activating FcγRIIA and inhibitory FcγRIIB[43]. Data from the FcγRIIA blocking experiments and Syk inhibition assays provide a pointer towards lack of participation of activating FcγRs including FcγRIIA in IVIG-induced β-catenin activation. These data thus suggest a possible role for FcγRIIB in inducing β-catenin activation although, we do not exclude the implication of yet another non-identified receptor. In fact, reports have demonstrated that interaction of IgG with certain receptors like FCRL5 (Fc receptor-like protein 5) on B cells requires intact IgG[3,4,75].

To explore the importance of β-catenin signaling in IVIG-mediated anti-inflammatory effects in vivo, we used EAE model. Our previous data show that IVIG has the ability to delay onset of the disease and to reduce the severity of EAE. The protection was associated with the inhibition of Th1/Th17 responses and peripheral expansion of Tregs[24,26]. Activation of the β-catenin pathway is also reported to suppress the severity of the EAE and to mediate reciprocal regulation of effector CD4⁺ T cells and Tregs[12]. EAE has been used by several groups to explore the anti-inflammatory mechanisms of IVIG. Although a recent report suggested possible neutralization of Mycobacterial antigens by F(ab′)₂ fragments as a mechanism of IVIG-mediated protection in EAE[76], other lines of evidences suggest that IVIG mechanisms in EAE go beyond mere neutralization of Mycobacterial antigens. For inducing EAE, Mycobacterial antigens are emulsified in Freund's adjuvant and then injected to the mice. Mycobacterial antigens in emulsion are not freely accessible to antibodies to get neutralized. It is contrary to the report of Quast et al.[76] who used free Mycobacterial antigens in ELISA to test binding of IVIG. Furthermore, other reports have shown the importance of Tregs, IL-11 receptor, IL-33 receptor in IVIG-mediated protection against EAE. Depletion of Tregs or deficiency of IL-11R, IL-33R, SIGN-R1 (Specific ICAM-3 grabbing nonintegrin-related 1), all led to abrogation of protection by IVIG[26,31,77,78].

Our data also advocate IgG structure-dependent diversity in the mechanisms of IVIG. While some mechanisms are dependent on F(ab′)₂ or Fc fragments[3–6,79], others like β-catenin activation as shown here requires intact IgG that were either not damaged or subjected to structural modifications. Moreover, the structural integrity of IgG is critical for the half-life of IVIG in the treated patients. Though both IVIG and β-catenin mediate similar actions, from the current perspective it is evident that IVIG does not link β-catenin to exert therapeutic benefits, as IVIG is able to reduce the severity of EAE and reciprocal regulation of Th1/Th17 and Treg responses independent of β-catenin. These data thus highlight that a single mechanism might not be entirely responsible for the sustained beneficial effects of IVIG in autoimmune and inflammatory diseases.

## Methods

**Reagents.** CD14 MicroBeads, basophil isolation kit II, rhGM-CSF, rhIL-4 were from Miltenyi Biotec (Paris, France). Rabbit MAbs to non-phospho (active) β-catenin (Ser33/37/Thr41) (clone D13A1, 1:500 dilution), GSK-3β (clone 27C10, 1:1000 dilution), GAPDH (HRP-conjugated) (clone 14C10, 1:1000 dilution), β-Actin (HRP-conjugated) (clone 13E5, 1:5000 dilution); polyclonal protein A and peptide affinity chromatography purified rabbit antibodies to phospho-β-catenin (Ser33/37/Thr41) (#9561, 1:500 dilution), β-catenin (#9562, 1:1000 dilution), mouse PCNA MAb (clone PC10, 1:1000 dilution) and HRP-conjugated affinity purified horse anti-mouse IgG secondary antibody (#7076, 1:2000 dilution) were from Cell Signaling Technology (Ozyme, Saint Quentin Yvelines, France). HRP-conjugated human adsorbed goat anti-rabbit IgG secondary antibody (#4010-05; 1:5000 dilution) was purchased from Southern Biotech (Birmingham, LA).

FITC-conjugated anti-mouse MAbs to IFN-γ (clone XMG1.2, 1:75 dilution for 0.1 million cells) and CD25 (clone 7D4, 1:30 dilution), anti-human CD32 (clone FLI8.26, 1:50 dilution), anti-human CD86 (clone FUN-1, 1:50 dilution); PE-conjugated anti-mouse MAb to IL-17A (clone TC11-18H10, 1:50 dilution); Per-CP/Cy5.5-conjugated anti-mouse MAb to CD4 (clone RM4-5, 1:50 dilution); APC/Cy7-conjugated anti-human MAb to CD69 (clone FN50, 1:100 dilution) were from BD Biosciences (Le Pont de Claix, France). APC-conjugated MAb to FOXP3 (clone FKJ-16s, 1:30 dilution), anti-human HLA-DR (clone G46-6, 1:50 dilution) and Fixable Viability Dye eFlour506 (1:500 dilution) were from eBioscience (Paris, France). Alexa Flour 488-conjugated goat anti-rabbit IgG (H + L) was from Invitrogen (ThermoFisher Scientific, Illkirch, France #A11034, 1:5000 dilution). Blocking MAb to FcγRIIA (clone IV.3) was purchased from Stem Cell Technologies (Grenoble, France).

Syk inhibitor R406 and β-Catenin/TCF Inhibitor, FH535 were obtained from InvivoGen (San Diego, CA) and Merck Chimie (Fontenay-sous-Bois, France) respectively. LPS (E. coli 055:B5) was purchased from Sigma-Aldrich (St. Quentin Fallavier, France).

Plasma-derived HSA and chitin were kind gifts from LFB Biomedicaments (Les Ulis, France) and Dr. V Aimanianda (Institut Pasteur, Paris, France). IL-3 was purchased from ImmunoTools (Friesoythe, Germany).

**Therapeutic normal IgG or IVIG.** As a source of IVIG, Privigen® and Hizentra® (CSL Behring) were used. IVIG was dialyzed against large volumes of RPMI 1640 at 4 °C for 4 h. For desialylation of IgG, IVIG was treated with recombinant neuraminidase (New England BioLabs, USA) as previously described[38,39]. Lectin-blot analyses were performed to validate the desialylation of IVIG (Supplementary Fig. 1). The sialylation content was measured by reverse phase high performance liquid chromatography[38,39].

F(ab′)₂ and Fc fragments of IVIG were prepared by pepsin and papain digestion respectively[56]. The F(ab′)₂ and Fc fragments were subjected to sodium dodecyl sulfate-polyacrylamide gel electrophoresis (SDS-PAGE) analyses for the verification of purity.

**Generation of human DCs.** Peripheral blood mononuclear cells (PBMCs) were obtained by subjecting the buffy coats of healthy donors to Ficoll density gradient centrifugation (Centre Necker-Cabanel, L'Établissement Français du Sang, Paris; EFS-INSERM ethical committee permission 15/EFS/012; 18/EFS/033). CD14⁺ cells were positively selected using CD14 MicroBeads (Miltenyi Biotec). Monocytes were cultured for 5 days in the presence of GM-CSF (1000 IU/10⁶ cells) and IL-4 (500 IU/10⁶ cells) to obtain DCs[80]. The gating strategy for DCs is provided in the Supplementary Fig. 2a.

**Induction of EAE in mice.** C57BL/6J female mice were obtained from Envigo laboratories, France. All the experiments on mice were done after approval from the ethical committee for animal experimentation and French Ministry of Higher Education and Research (APAFIS#10539-2017070715163055 V4). EAE was

induced in 8-week-old mice as described in our earlier studies[24,26]. Briefly, mice were immunized on day 0 by subcutaneously injecting 200 μg of MOG$_{35-55}$ peptide (MEVGWYRSPFSRVVHLYRNGK; synthesized at Polypeptide group, Strasbourg, France; 95% purity) emulsified in complete Freund's adjuvant, containing 880 μg of killed *Mycobacterium tuberculosis* strain H37RA. Further, all mice received 300 ng pertussis toxin on day 0 and day 2. Mice were observed every day for the development of clinical signs of EAE. Clinical signs in control mice typically appeared between 10–12 days post immunization. Mice were scored based on the following scoring system where, 0, no signs; 1, tail paresis; 2, hind limb paresis; 3, hind limb paralysis; 4, tetraplegia; and 5, moribund. Any animal reached the score 4 was euthanized for ethical reasons.

Mice were divided into three groups. Control group, IVIG + DMSO group and IVIG + inhibitor group. Mice received by intraperitoneal route, 0.8 g/kg of IVIG in combination with either 50 μl of DMSO (Sigma-Aldrich) or 15 mg/kg of β-catenin inhibitor (FH535) every day until peak of the disease.

**Isolation of mouse splenocytes and CD4$^+$ T cell analysis**. On the day of onset of the disease, mice were sacrificed to collect spleen. Splenocytes were isolated and red blood cells were lysed using ACK (Ammonium-Chloride-Potassium) lysis buffer. Cells (0.5 million/ml) were stimulated with phorbol myristate acetate (PMA) (50 ng/ml/0.5 million cells) and ionomycin (500 ng/ml/0.5 million cells), along with GolgiStop for 4 h. Th1, Th17, and Treg populations were analyzed by combination of surface and intracellular staining for various markers. Surface staining was performed with fluorescence-conjugated MAbs to CD4 and CD25. After fixation and permeabilization by intracellular staining kit (eBioscience), intracellular staining with fluorescence-conjugated MAbs to IFN-γ, IL-17A, and FOXP3 were carried out. Samples were acquired using LSR-II flow cytometer (BD Biosciences) and data were analyzed by FACS-DIVA (BD Biosciences). The gating strategy for CD4$^+$ T cells is provided in the Supplementary Fig. 2b.

**Treatment of DCs and mice splenocytes**. DCs (0.5 million cells/ml) were cultured in RPMI-1640 containing 10% fetal calf serum (FCS) alone or with 25 mg/ml or 15 mg/ml of IVIG or desialylated IVIG (25 mg/ml) or equimolar concentration of F(ab')$_2$ fragments, Fc fragments or HSA (irrelevant protein control) for 24 h. Activation of the β-catenin signaling was analyzed by immunoblotting.

For the experiments involving treatment of pharmacological inhibitors, concentrations were chosen after titration. β-Catenin/TCF inhibitor, FH535 was used to inhibit β-Catenin at the concentration of 15 μM, 2 h prior to the treatment of IVIG.

In other experiments, DCs were pretreated with pre-determined concentrations of EDTA (0.5 mM) or with Syk inhibitor R406 (10 μM) or with blocking MAb to FcγRIIA (10 μg/0.5 million cells) for one hour before culturing with IVIG. We confirmed that aforementioned concentrations of EDTA, Syk inhibitor and FcγRIIA MAb were functional (Supplementary Fig. 3).

**RNA isolation and quantitative RT-PCR**. For the analyses of gene expression, cells were treated with IVIG (25 mg/ml) or HSA for 12 h. Untreated cells were used as control. Total RNA from the different conditions was isolated using the RNeasy Mini Kit (Qiagen, Hilden, Germany). cDNA was synthesized using High-Capacity cDNA Reverse Transcription kit (ThermoFisher Scientific, Courtaboeuf, France). Quantitative RT-PCR for *CTNNB1* was done by TaqMan Universal Master Mix II with UNG (Applied Biosystems, Foster City, CA), and expression was measured with TaqMan Gene Expression Assays (Applied Biosystems, Hs00355045_m1 (CTNNB1) and Hs02786624_g1 (glyceraldehyde 3-phosphate dehydrogenase, GAPDH).

For the analyses of expression of *Wnt* genes, SYBR$^{TM}$ Green PCR Master Mix (ThermoFisher Scientific) was used. Gene expression was calculated relative to the housekeeping gene *GAPDH*. The primers used in the study are as follows:

*WNT1*- F: CAGCGACAACATTGACTTCG,
R: GCCTCGTTGTTGTGAAGGTT;
*WNT3A*- F: CCTGCACTCCATCCAGCTACA,
R: GACCTCTCTTCCTACCTTTCCCTTA;
*WNT5A*- F: CTCACTGAAATGCGTGTTGG,
R: AATGCCCTCTCCACAAAGTG;
*WNT7A*- F: CCCACCTTCCTGAAGATCAA,
R: GTCCTCCTCGCAGTAGTTGG;
*WNT7B*- F: GGCACAAGGACCTACCAGAG,
R: CCTGATGTGTTCTCCCAGGT;
*WNT10A*- F: CCACTCCGACCTGGTCTACTTTG,
R: TGCTGCTCTTATTGCACAGGC;
*GAPDH*- F: CGACCACTTTGTCAAGCTCA,
R: GGTGGTCCAGGGGTCTTACT.

**Transfection with siRNA**. LRP5 and LRP6 predesigned Accell SMART Pool siRNA was purchased from Dharmacon (Thermo Fisher Scientific) and introduced into cells according to manufacturer's protocol. Briefly, DCs were cultured in 12 well plate at 0.5 × 10$^6$ cells/0.5 ml of Accell Delivery media. One micrometer of nontargeting control siRNA or LRP5 and LRP6 siRNA were introduced into cells for 72 h. After 72 h, cells were cultured in RPMI with 10% FCS and IVIG for 24 h

followed by FACS analysis. Surface staining of DC was performed with fluorescence-conjugated MAb to HLA-DR. For active-β-catenin detection, cells were stained with rabbit MAb to non-phospho (active) β-catenin (Ser33/37/Thr41) and followed by Alexa Flour 488-conjugated goat anti-rabbit IgG (H + L) by using Cell Signaling Buffer Set A (Miltenyi Biotec).

**Immunoblotting**. Total cell lysates were obtained by lysing cells in RIPA buffer (radioimmunoprecipitation assay buffer, Sigma-Aldrich) containing protease and phosphatase inhibitors (Sigma-Aldrich). Proteins were quantified using Bio-Rad protein assay reagent (Bio-Rad, Marnes-la-Coquette, France) and an equal amount of proteins were resolved on SDS-PAGE and transferred on to nitrocellulose, iBlot$^{TM}$ Transfer Stack (ThermoFisher Scientific). The membrane was blocked using 5% bovine serum albumin (BSA) in TBST (Tris-buffered saline (TBS) and Polysorbate 20 or Tween 20) for 60 min. The blots were incubated overnight at 4 °C with various primary antibodies diluted in TBST-5% BSA followed by incubation with secondary antibody for 2 h. After washing blots in TBST thrice with 10 min interval, blots were developed with SuperSignal$^{TM}$ WestDura Extended Duration substrate (Thermo Fisher Scientific). β-actin or GAPDH were used as a loading control. The western blot images were analyzed by myImageAnalysis software v 2.0 (Thermo Fisher Scientific). The full western blot images are provided in the Supplementary Fig. 4.

**Analyses of nuclear translocation of β-catenin**. DCs were cultured alone or in presence of IVIG or equimolar concentration of HSA for 24 h. Cells were harvested and pellets were washed twice with ice cold PBS and resuspended in hypotonic lysis buffer (20 mM Tris pH 7.5, 5 mM MgCl$_2$, 1 mM EDTA, and 240 mM Sucrose). After incubating for 10 min on ice, cells were centrifuged at 13,000 rpm for 10 min at 4 °C to pellet down the nuclei. Supernatant was collected and used as a cytosolic extract. Pellets were resuspended in ice cold Buffer C (20 % Glycerol, 20 mM HEPES pH7.9, 420 mM NaCl, 1.5 mM MgCl$_2$ and 0.2 mM EDTA). Pellets were incubated on ice for 30 min followed by centrifugation at 13,000 rpm for 20 min at 4 °C to collect the nuclear extract. Immunoblotting of cytosolic and nuclear extracts was performed as detailed above.

**ELISA for measuring the cytokines and Wnt ligands**. DCs were stimulated with LPS (100 ng/0.5 million cells /ml) for 24 h. The cells were hen treated with β-catenin antagonist FH535 for 2 h followed by culture with IVIG (25 mg/ml) for additional 24 h. Cell-free supernatants were analyzed for the secretion of IL-8 and IL-6 by ELISA (ELISA Ready-SET-Go, eBioscience).

Secretion of Wnt3a and Wnt5a was analyzed in cell-free supernatants from IVIG-treated DCs with the help of Human Protein Wnt-3a and Wnt-5a ELISA Kit (MyBioSource, San Diego, CA).

**Statistical analyses**. As highlighted in the figure legends, the experiments were repeated several times by using independent donors or mice to ensure reproducibility. Human data were from three to seven donors except for nuclear translocation of β-catenin that was representative of two donors. Mice data were from two independent experiments. Statistical analyses were performed by using one-way ANOVA followed by Tukey's multiple comparisons test or by two-way ANOVA with Bonferroni post-t-test for in vivo experiments. $P < 0.05$ was considered significant.

**Reporting summary**. Further information on research design is available in the Nature Research Reporting Summary linked to this article.

## Data availability
The authors declare that all data supporting the findings of this study are available within the paper and its supplementary information. Source data underlying the graphs presented in the main figures is available in Supplementary Data 1

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

## Acknowledgements

Supported by Institut National de la Santé et de la Recherche Médicale, Sorbonne Université and Université Paris Descartes, France; A.K. was a recipient of fellowship from Indo-French Center for Promotion of Advanced Research (CEFIPRA) and La Fondation pour la Recherche Médicale (FDT201805005552). M.B.J. was a recipient of a fellowship from Ministère de l'enseignement supérieur et de la recherche (Ecole doctorale 394, Sorbonne Université, France). We thank Dr. C Galeotti and Dr. E Stephen-Victor for the support. We also thank the staff of core facilities at Centre de Recherche des Cordeliers for the help: Centre d'Histologie, d'Imagerie et de Cyto-métrie (CHIC), Centre d'Explorations Fonctionnelles (CEF), and Le Centre de Gén-otypage et de Biochimie (CGB).

## Author contributions

A.K., N.R., M.D., M.B.J., S.D., and J.B. performed experiments; A.K., S.L.-D., S.V.K., and J.B. analyzed experimental data; F.K. performed desialylation of IVIG; A.K. and J.B. wrote the paper; all authors revised and approved the manuscript critically for important intellectual content and approved the final version.

## Competing interests

F.K. is an employee of CSL Behring AG, Switzerland. Other authors have no relevant competing interests.
