## [Peer Review File · Communications Biology]

Reviewers' comments:

Reviewer #1 (Remarks to the Author):

Karnam et al. investigated whether IVIG induces the β -catenin pathway to exert its anti-inflammatory effects. They report that IVIG activates the β -catenin γ in human DC an IgG-sialylation independent manner. β -catenin activation by IVIG requires LRP5/6 co-receptors but $Fc\gamma RII$ and Syk pathways are not implicated. Induction of the β -catenin pathway is not sufficient to reciprocally regulate effector Th17/Th1 and regulatory T cells after EAE induction and does not protect against EAE.

Overall the study is well performed and well written. The impact of the major finding, induction of β -catenin signalling by IVIG, is limited due to the observation that inhibition of Wnt/ β -catenin signaling had no effect on anti-inflammatory properties of IVIG in vitro and in vivo. The latter indicates that this pathway is dispensable for the anti inflammatory action of IVIG, as discussed by the authors.

Other points:

1. The ms could be improved by showing whether induction of the β -catenin pathway is mediated by F(ab)2 vs. Fc fragments.

Figure 3: please, confirm the efficacy of desialylation (e.g. by lectin-blot)

Figure 4: Inhibitory compounds (EDTA, anti-CD32, Syk inhibitor) should be titrated and/or a positive control for the inhibitory efficacy should be provided.

Figure 4: Does the CD32-targeting antibody specifically bind to $Fc\gamma RIIB$? Please discuss.

Figure 8: An additional in vivo autoimmune disease model should be tested to investigate the protective effect of IVIG and its inhibition by β -catenin antagonists. Studies reported that the protective effect of IVIG in EAE is lost in the adoptive transfer model of EAE and is likely driven by IgG species neutralizing microbial antigens which are required to induce active EAE (e.g. Quast et al., J Neuroinflamm 2016).

Reviewer #2 (Remarks to the Author):

In this report, Karnam et al explored the anti-inflammatory pathway of intravenous immunoglobulin, one of the most commonly used immunotherapies for autoimmune and inflammatory diseases. They report that intravenous immunoglobulin activates the β -catenin pathway along with Wnt5a secretion in human dendritic cells in a sialylation independent manner. Mechanistically, they found that β -catenin activation by intravenous immunoglobulin requires LRP5/6 co-receptors but receptors for Fc and Syk pathways are not implicated. Despite induction of canonical β -catenin/Wnt signaling, this pathway appeared to be dispensable for the anti-inflammatory action of intravenous immunoglobulin both in vitro, and in vivo.

Overall, authors have performed the experiments in a diligent manner and their data support the conclusion.

1. It will be worth, if authors show intra nuclear translocation of β -catenin when dendritic cells are treated with intravenous immunoglobulin. This will further validate their findings

2. The role of sialylation of IgG: Many studies have shown that sialylation of Fc fragment is important for the anti-inflammatory actions of intravenous immunoglobulin. Although sialylation of IgG was found to be dispensable for the induction of β -catenin activation, confirming this with isolated Fc fragments of intravenous immunoglobulin would further strengthen the data.

Reviewer #1 (Remarks to the Author):

Karnam et al. investigated whether IVIG induces the β -catenin pathway to exert its anti-inflammatory effects. They report that IVIG activates the β -catenin in human DC in an IgG-sialylation independent manner. β -catenin activation by IVIG requires LRP5/6 co-receptors but Fc γ R2 and Syk pathways are not implicated. Induction of the β -catenin pathway is not sufficient to reciprocally regulate effector Th17/Th1 and regulatory T cells after EAE induction and does not protect against EAE.

Overall the study is well performed and well written. The impact of the major finding, induction of β -catenin signalling by IVIG, is limited due to the observation that inhibition of Wnt/ β -catenin signaling had no effect on anti-inflammatory properties of IVIG in vitro and in vivo. The latter indicates that this pathway is dispensable for the anti-inflammatory action of IVIG, as discussed by the authors.

Other points:

1. The ms could be improved by showing whether induction of the β -catenin pathway is mediated by F(ab)₂ vs. Fc fragments.

Response) Thank you for the excellent suggestion. We performed the experiments as suggested by the reviewer. We however found that intact IgG is mandatory for the β -catenin activation by intravenous immunoglobulin.

DCs were treated with IVIG (25 mg/ml) and equimolar concentrations of F(ab')₂ or Fc fragments. Consistent with the data on the lack of implications of sialylation of IgG, Syk and Fc γ R2A in mediating β -catenin activation by IVIG, Fc fragments of IVIG failed to induce β -catenin activation (Fig 5a).

These results raised the prospect that IVIG might induce β -catenin activation via F(ab')₂ fragments. However, we did not observe β -catenin activation even with F(ab')₂ fragments of IVIG (Fig 5b). Together our data indicate that intact IgG is mandatory for the β -catenin activation by IVIG.

We discussed these results in the discussion section of the article.

“We found that induction of β -catenin activation by IVIG requires intact IgG while neither Fc nor F(ab')₂ fragments could recapitulate these functions. As human monocyte-derived DC mainly express low affinity Fc γ R2, explains the inability of Fc fragments of IVIG to induce β -catenin activation. However, despite having the ability to bind DC¹⁹, F(ab')₂ fragments did not promote β -catenin activation. Therefore, it is likely that cooperation between Fab and Fc regions lead to β -catenin activation by IVIG. As reported earlier, IVIG contains antibodies to various self-motifs including those antigens expressed on DC4-6. Thus, binding of Fab region of IVIG to DC might license Fc region to bind Fc γ R2 of adjacent DC and signal β -catenin activation in those cells. Human DC display a balanced expression of activating Fc γ R2A and inhibitory Fc γ R2B⁴³. Data from the Fc γ R2A blocking experiments and Syk inhibition assays

provide a pointer towards lack of participation of activating FcγRs including FcγRIIA in IVIG-induced β-catenin activation. These data thus suggest a possible role for FcγRIIB in inducing β-catenin activation although, we do not exclude the implication of yet another non-identified receptor. In fact, reports have demonstrated that interaction of IgG with certain receptors like FCRL5 (Fc receptor-like protein 5) on B cells requires intact IgG^{3,4,75}.”

“Our data also advocate IgG structure-dependent diversity in the mechanisms of IVIG. While some mechanisms are dependent on F(ab')₂ or Fc fragments³⁻⁶, others like β-catenin activation as shown here requires intact IgG that were either not damaged or subjected to structural modifications. Moreover, the structural integrity of IgG is critical for the half-life of IVIG in the treated patients.”

Figure 3: please, confirm the efficacy of desialylation (e.g. by lectin-blot)

Response) We confirmed the efficacy of desialylation by lectin blot (Supplementary Figure 1). As we used same desialylated IVIG that was used in our previous publication (*Eur. J. Immunol.* 44, 2059-2063, (2014), we are reproducing the image with modifications by permission from John Wiley and Sons, Inc.

Figure 4: Inhibitory compounds (EDTA, anti-CD32, Syk inhibitor) should be titrated and/or a positive control for the inhibitory efficacy should be provided.

Response) We have provided positive controls of the inhibitory compounds (EDTA, anti-CD32, Syk inhibitor) in **Supplementary Figure 3**. These data validate the inhibitory capacity of various reagents.

Figure 4: Does the CD32-targeting antibody specifically bind to FcγRIIB ? Please discuss.

Response) CD32-targeting antibody specifically binds to FcγRIIA. We have clarified this in the revised manuscript. The details are provided in the Results (page 7) and Materials and Methods page 21, paragraph 4). The data are also discussed in page 15 (second paragraph)

Figure 8: An additional in vivo autoimmune disease model should be tested to investigate the protective effect of IVIG and its inhibition by β-catenin antagonists. Studies reported that the protective effect of IVIG in EAE is lost in the adoptive transfer model of EAE and is likely driven by IgG species neutralizing microbial antigens which are required to induce active EAE (e.g. Quast et al., J Neuroinflamm 2016).

Response) We although respect the view of the reviewer, we feel that we might not get ethical permission for mere confirming the protective effect of IVIG and its inhibition by β-catenin antagonists in another model. The main reason is that it will be redundant to what is already shown in the EAE model and hence violates 3R rule of animal experimentation.

Also establishing a new model would take considerable amount of time. The main idea of our article is not to demonstrate that IVIG benefits autoimmune and inflammatory diseases. It is already demonstrated in several experimental models and in large number of autoimmune and inflammatory diseases of human (data from around the world and our own data). Our investigation aims at how does IVIG benefit autoimmune pathologies and what are the underlying mechanisms. In this context, we have shown that both in **HUMAN immune cells** and in mice, IVIG induces β -catenin pathway and in both the systems, this pathway is dispensable. Thus, irrespective of animal model and more importantly in humans, we have obtained similar results thus validating our findings.

Therefore, we have discussed the points raised by the reviewer in the revised manuscript (page 16)

“EAE has been used by several groups to explore the anti-inflammatory mechanisms of IVIG. Although a recent report suggested possible neutralization of Mycobacterial antigens by F(ab')₂ fragments as a mechanism of IVIG-mediated protection in EAE⁷⁶, other lines of evidences suggest that IVIG mechanisms in EAE go beyond mere neutralization of Mycobacterial antigens. For inducing EAE, Mycobacterial antigens are emulsified in Freund’s adjuvant and then injected to the mice. Mycobacterial antigens in emulsion are not freely accessible to antibodies to get neutralized. It is contrary to the report of Quast et al. who used free Mycobacterial antigens in ELISA to test binding of IVIG⁷⁶. Furthermore, other reports have shown the importance of Tregs, IL-11 receptor, IL-33 receptor in IVIG-mediated protection against EAE. Depletion of Tregs or deficiency of IL-11R, IL-33R, SIGN-R1 (Specific ICAM-3 grabbing nonintegrin-related 1), all led to abrogation of protection by IVIG^{26,31,77,78}.”

Reviewer #2 (Remarks to the Author):

In this report, Karnam et al explored the anti-inflammatory pathway of intravenous immunoglobulin, one of the most commonly used immunotherapies for autoimmune and inflammatory diseases. Thy report that intravenous immunoglobulin activates the β -catenin pathway along with Wnt5a secretion in human dendritic cells in a sialylation independent manner. Mechanistically, they found that β -catenin activation by intravenous immunoglobulin requires LRP5/6 co-receptors but receptors for Fc and Syk pathways are not implicated. Despite induction of canonical β -catenin/Wnt signaling, this pathway was appeared to be dispensable for the anti-inflammatory action of intravenous immunoglobulin both in vitro, and in vivo.

Overall, authors have performed the experiments in a diligent manner and their data support the conclusion.

1. It will be worth, if authors show intra nuclear translocation of β -catenin when dendritic cells are treated with intravenous immunoglobulin. This will further validate their findings

Response) Thank you for the excellent suggestion. As suggested by the reviewer, we investigated intra nuclear translocation of β -catenin when dendritic cells are treated with

intravenous immunoglobulin (Figure 1c). Representative immunoblot of two donors is presented.

2. The role of sialylation of IgG: Many studies have shown that sialylation of Fc fragment is important for the anti-inflammatory actions of intravenous immunoglobulin. Although sialylation of IgG was found to be dispensable for the induction of β -catenin activation, confirming this with isolated Fc fragments of intravenous immunoglobulin would further strengthen the data.

Response) This question partly overlaps with that of first reviewer. Thank you again for the excellent suggestion. We however found that intact IgG is mandatory for the β -catenin activation by intravenous immunoglobulin.

DCs were treated with IVIG (25 mg/ml) and equimolar concentrations of F(ab')₂ or Fc fragments. Consistent with the data on the lack of implications of sialylation of IgG, Syk and Fc γ RIIA in mediating β -catenin activation by IVIG, Fc fragments of IVIG failed to induce β -catenin activation (Fig 5a).

These results raised the prospect that IVIG might induce β -catenin activation via F(ab')₂ fragments. However, we did not observe β -catenin activation even with F(ab')₂ fragments of IVIG (Fig 5b). Together our data indicate that intact IgG is mandatory for the β -catenin activation by IVIG.

We discussed these results in the discussion section of the article.

“We found that induction of β -catenin activation by IVIG requires intact IgG while neither Fc nor F(ab')₂ fragments could recapitulate these functions. As human monocyte-derived DC mainly express low affinity Fc γ RII, explains the inability of Fc fragments of IVIG to induce β -catenin activation. However, despite having the ability to bind DC¹⁹, F(ab')₂ fragments did not promote β -catenin activation. Therefore, it is likely that cooperation between Fab and Fc regions lead to β -catenin activation by IVIG. As reported earlier, IVIG contains antibodies to various self-motifs including those antigens expressed on DC4-6. Thus, binding of Fab region of IVIG to DC might license Fc region to bind Fc γ RII of adjacent DC and signal β -catenin activation in those cells. Human DC display a balanced expression of activating Fc γ RIIA and inhibitory Fc γ RIIB⁴³. Data from the Fc γ RIIA blocking experiments and Syk inhibition assays provide a pointer towards lack of participation of activating Fc γ Rs including Fc γ RIIA in IVIG-induced β -catenin activation. These data thus suggest a possible role for Fc γ RIIB in inducing β -catenin activation although, we do not exclude the implication of yet another non-identified receptor. In fact, reports have demonstrated that interaction of IgG with certain receptors like FCRL5 (Fc receptor-like protein 5) on B cells requires intact IgG^{3,4,75}.”

“Our data also advocate IgG structure-dependent diversity in the mechanisms of IVIG. While some mechanisms are dependent on F(ab')₂ or Fc fragments³⁻⁶, others like β -catenin activation as shown here requires intact IgG that were either not damaged or subjected to structural modifications. Moreover, the structural integrity of IgG is critical for the half-life of IVIG in the treated patients.”

REVIEWERS' COMMENTS:

Reviewer #1 (Remarks to the Author):

The authors adequately addressed most of the points raised by this reviewer. No major concerns remain.

Reviewer #2 (Remarks to the Author):

The authors addressed all my questions